# Ccl2/Ccr2 signalling recruits a distinct fetal microchimeric population that rescues delayed maternal wound healing

Mathieu Castela[1,2], Dany Nassar[3,4,*], Maria Sbeih[1,2,*], Marie Jachiet[1,2,3], Zhe Wang[1,**] & Selim Aractingi[1,3,5,**]

Foetal microchimeric cells (FMCs) traffic into maternal circulation during pregnancy and persist for decades after delivery. Upon maternal injury, FMCs migrate to affected sites where they participate in tissue healing. However, the specific signals regulating the trafficking of FMCs to injury sites had to be identified. Here we report that, in mice, a subset of FMCs implicated in tissue repair displays CD11b$^+$ CD34$^+$ CD31$^+$ phenotype and highly express C-C chemokine receptor 2 (Ccr2). The Ccr2 ligand chemokine ligand 2 (Ccl2) enhances the recruitment of FMCs to maternal wounds where these cells transdifferentiate into endothelial cells and stimulate angiogenesis through Cxcl1 secretion. Ccl2 administration improves delayed maternal wound healing in pregnant and postpartum mice but never in virgin ones. This role of Ccl2/Ccr2 signalling opens new strategies for tissue repair through natural stem cell therapy, a concept that can be later applied to other types of maternal diseases.

[1] INSERM UMRS_938, Saint-Antoine Research Center, 27, rue de Chaligny, Paris 75012, France. [2] UPMC Université Paris 6, 4, place Jussier, Paris 75005, France. [3] Université Paris 5 Descartes, 12, rue de l'Ecole de Médecine, Paris 75006, France. [4] Department of Dermatology, American University of Beirut, Riad El Soph, Beinut 11072020, Lebanon. [5] Department of Dermatology, Hôpital Cochin, AP-HP, 89, rue d'Assas, Paris 75006, France. * These authors contributed equally to this work. ** These authors jointly supervised this work. Correspondence and requests for materials should be addressed to S.A. (email: selim.aractingi@gmail.com).

Foetal microchimeric cells (FMCs) enter maternal circulation, engraft into various organs and persist for decades after delivery, probably for the mother's remaining lifespan[1]. Foetal cells nest in maternal bone marrow (BM) and remain well tolerated by the maternal immune system[1]. FMCs have been reported to contain progenitor cells, including lymphoid[2], hematopoietic[3–5], mesenchymal[6,7] and cardiomyocyte[8] progenitors. Upon various types of maternal injury, these cells can be triggered to migrate to various maternal tissues in which they will adopt the phenotype of the damaged organ[4,9]. Indeed, FMCs are able to differentiate into neurons[10], hepatocytes[11] endothelial cells[12,13], thyroid, cervix and gut[9,14] cells in concerned maternal tissues.

Wound healing is an intricate, interactive biological process[15]. There is a well-coordinated interplay between tissue structures and residential cells as well as distant cells from BM that are essential for the healing process[16]. Studies using chimeric mice with tagged BM identified various types of BM-derived endothelial progenitor cells (EPCs) participating in wound angiogenesis[17,18]. The first reported one is a $CD34^+$ $VEGFR2^+/CD31^+$ $CD45^-$ population that can form blood vessels[19–21]. Interestingly, others identified a $CD34^+$ $VEGFR2^+/CD31^+$ $CD45^+$ or $CD11b^+$ population from the BM that secretes VEGF-A and indirectly supports wound angiogenesis[22]. However, more recent studies have finally failed to detect BM-derived EPCs in healing adult tissues[22,23].

We and others have reported the constant presence of fetal cells in maternal wounds during and after delivery[13,24], including chronic wounds. In maternal wound beds, fetal cells have been shown to differentiate into endothelial cells, able to form full blood vessels, connected to maternal circulation, in such wounds[13]. The fetal cells mobilized in response to maternal wounds did not express the classically reported EPC markers $CD34^+$, $CD11b^-$ and $VEGFR2^+$ (ref. 13). We have also detected blood and lymphatic endothelial cells of fetal origin in inflamed skin[12] and in melanomas[25]. Thus fetal progenitors constitute a reservoir of progenitors, well tolerated by the maternal immune system, that are involved in maternal neovascularization during wound healing[1,13].

Here we investigate the molecular mechanisms that selectively recruit fetal cells into maternal skin wounds in mice. We find that C-C chemokine receptor 2 (Ccr2) was overexpressed on FMCs upon maternal wounding. Ccr2 is a chemokine receptor expressed on BM progenitor cells as well as monocytes[26,27]. Chemokine ligand 2 (Ccl2), its main ligand, is secreted by endothelial cells and macrophages[28,29]. It triggers the recruitment of $Ccr2^+$ various cell types from the marrow to the secreting site. Here Ccl2, given at physiological doses, was able to induce maternal neoangiogenesis and to improve skin healing by recruiting FMCs. Importantly, such effect of Ccl2 was restricted to pregnant and postpartum mice and was dependent on Ccr2 expression on FMCs. Our results pave the way for natural stem cell therapy for tissue repair in diseased females.

## Results

**Maternal skin wounding recruits FMCs through Ccr2 signalling.** We mated virgin C57BL/6 females with males heterozygous with the enhanced green fluorescence protein (eGFP) gene. Excisional wounds were then performed on dorsal skin at day E15.5 (experimental scheme is described in detail in Supplementary Fig. 1). One day after skin wounding, fetal cell counts rapidly increased in the BM (Fig. 1a), blood (Fig. 1b) and wound bed (Fig. 1c). The abundance of these cells in BM and blood decreased on day 3 but were still detectable at low level on days 5 and 9, indicating that FMC mobilization in these compartments was an

early process. Details of sensitivity and specificity about our cell sorting experiments are given in the Methods section and Supplementary Fig. 2. To identify the signalling pathways that mediate the mobilization of fetal cells upon maternal wounding, we performed PCR array analysis on sorted fetal cells from the BM isolated from pregnant mice with and without skin wounds. Ccr2 was found to be the chemokine receptor displaying the highest upregulation upon wounding (Fig. 1d and Supplementary Data 1). We performed also RNA sequencing on sorted fetal $eGFP^+$ cells from the peripheral blood obtained from wounded and unwounded pregnant mice. Since fetal cells show a very early peak in the blood (at day 1) after wounding, we first selected genes that displayed high expression ($>100$ count per million) in wounded mothers. Then we selected the genes that were over-expressed in circulating fetal cells from wounded mice as compared to unwounded mice ($>1.25$-fold change). Using these thresholds, we retrieved a list of 558 genes that were over-expressed in circulating fetal cells from wounded mice. Among these genes, 16 corresponded to surface receptors and the only chemokine receptor was Ccr2 with 376 versus 261 counts per million reads in wounded versus unwounded conditions leading to a fold change of 1.44 (Fig. 1e). Accordingly, we checked the Ccr2 expression of fetal cells in maternal wounds. By immuno-fluorescence analysis, FMCs expressed Ccr2 in wound tissue (Fig. 1f). Moreover, 90% of the circulating fetal cells expressed Ccr2 1 day after skin injury versus only 1.50% of those from pregnant unwounded mice (Fig. 1g). These results suggest that maternal wounding induces an early expansion of the circulating $Ccr2^+$ fetal cell population.

**Monocytes secrete Ccl2 early in skin wound healing.** We next investigated the Ccr2-mediated chemotactic cues in skin wounds. Ccr2 mRNA and protein levels increased strongly in skin wounds 1 day after injury, subsequently decreasing on day 3 (Fig. 1h,k). Ccl2 mRNA levels followed a parallel pattern, increasing on day 1 then decreasing on day 3 (Fig. 1i). Meanwhile, the levels of Ccl8, another Ccr2 ligand, peaked later in the wound, on day 3 (Fig. 1j). Thus Ccl2/Ccr2 signalling corresponds with the observed early peak of postwounding fetal cells.

We investigated the specific populations secreting Ccl2 in skin wounds by measuring Ccl2 levels in various types of cells sorted from the wounds. Marked Ccl2 overexpression was detected in $CD45^+$ leukocytes from day 1 wounds (Fig. 1m). We carried out double immunofluorescence staining on day 1 wound sections on day 1: $F4/80^+$ monocytes and $CD31^+$ endothelial cells expressed Ccl2, whereas $GR-1^+$ neutrophils did not (Fig. 1l). These data suggest that monocytes and endothelial cells secrete Ccl2 during the initial stage of skin wound healing.

**Ccl2 recruits fetal cells to maternal wounds.** Given the over-expression of Ccr2 in FMCs upon wounding, the high percentage of FMCs expressing Ccr2 on their surface and the overexpression of Ccl2 in wound tissue during early stages of healing, we hypothesized that Ccl2 mediates the early recruitment of FMCs to wounds. We therefore injected physiological dose of Ccl2 (50 ng) or PBS into the wound bed on days 0 and 2 after wounding in E15.5 pregnant or virgin mice (Fig. 2a). The numbers of $eGFP^+$ cells in blood did not differ between the mice receiving Ccl2 and PBS on day 7 (Supplementary Fig. 3a). In contrast, there were 2.03-folds more $eGFP^+$ cells in skin wound tissues into which Ccl2 had been injected as in control wounds (Fig. 2b,c). Consistent with our FACS results, by performing immuno-fluorescence staining, we detected 3.38-folds more as many $eGFP^+$ cells in the sections of mice receiving Ccl2 as in those

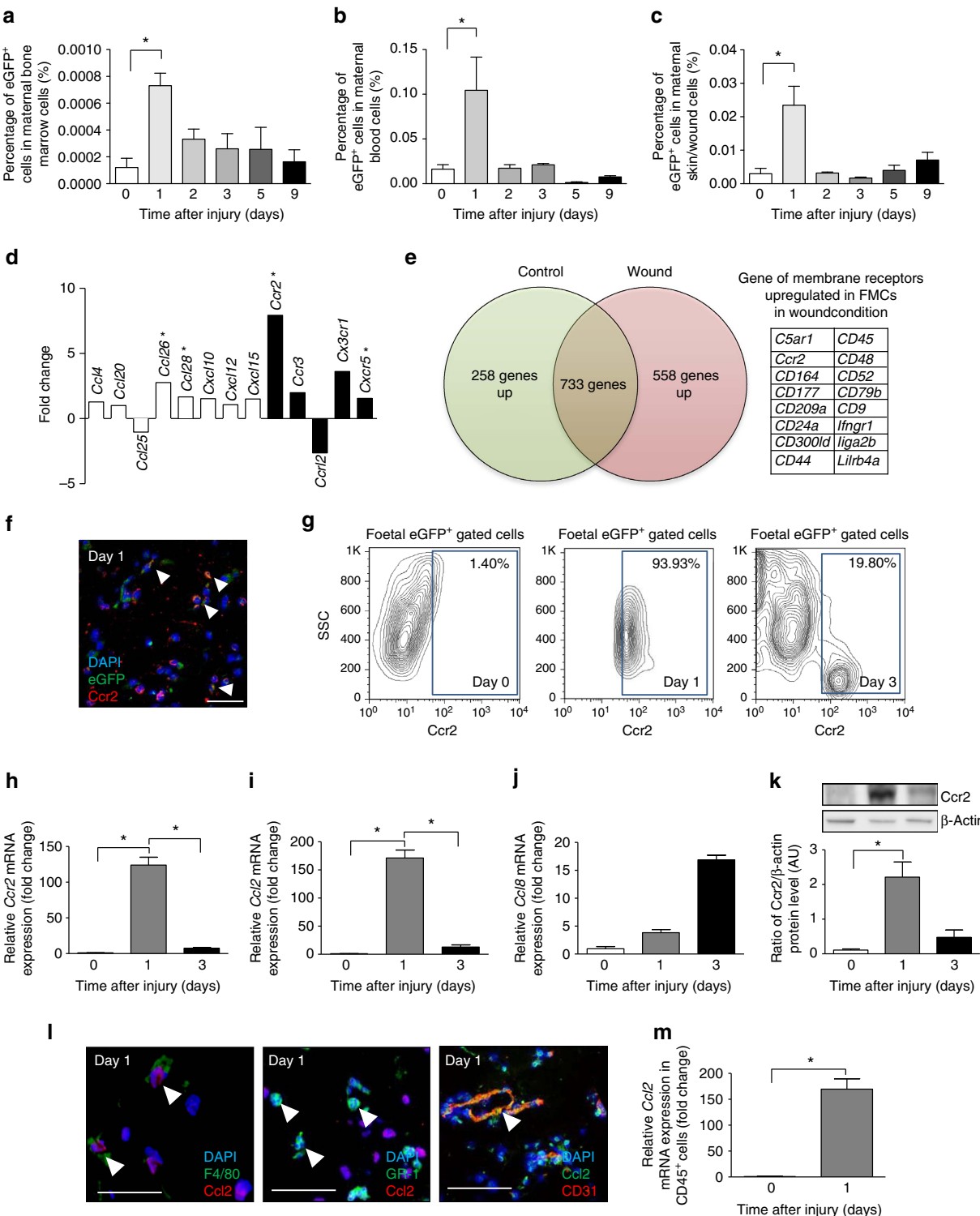

**Figure 1 | Maternal wounding activates FMCs and induces Ccr2 production.** Quantification by FACS of fetal cells in the bone marrow (**a**), blood (**b**) and skin/wound (**c**) after maternal skin injury ($n = 4$-9). (**d**) PCR array analysis of cytokine and chemokine gene expression in FMCs sorted from maternal bone marrow in mice with or without wounds on day 3 ($n = 6$). White bars represent the ligand genes and black bars represent the receptor genes. (**e**) RNA sequencing of circulating FMCs in maternal blood during pregnancy with or without wounds. (**f**) Labelling for Ccr2 (red) and natural eGFP (green) fluorescence of the wounds (day 1 after wounding) of pregnant female mice carrying eGFP$^+$ foetuses. White arrowheads indicate colocalization. Scale bars: 50 μm. (**g**) FACS analysis of Ccr2$^+$ cells among eGFP$^+$ FMCs in the peripheral blood, with and without maternal wounding ($n = 3$). Quantitative RT–PCR analysis for Ccr2 (**h**) and its ligand Ccl2 (**i**) and Ccl8 (**j**). Levels of mRNA normalized against mRNA levels for Gapdh in normal skin and wounds ($n = 3$). (**k**) Western blot of Ccr2 in normal skin and wounds at differents time points. Representative results from two indepedent experiments are shown. (**l**) Dual labelling for Ccl2 (red) and F4/80 (green) in the wound on day 1. White arrowheads indicate colocalization. Dual labelling for GR-1 (green) and Ccl2 (red) in a wound on day 1, with white arrowheads showing single staining for GR-1. Dual labelling for Ccl2 (green) and CD31 (red) in the wound. Scale bars: 50 μm. (**m**) Quantitative RT–PCR analysis for Ccl2 in sorted leukocytes from day 1 wounds. Student's t-test, *$P < 0.05$; mean ± s.e.m.

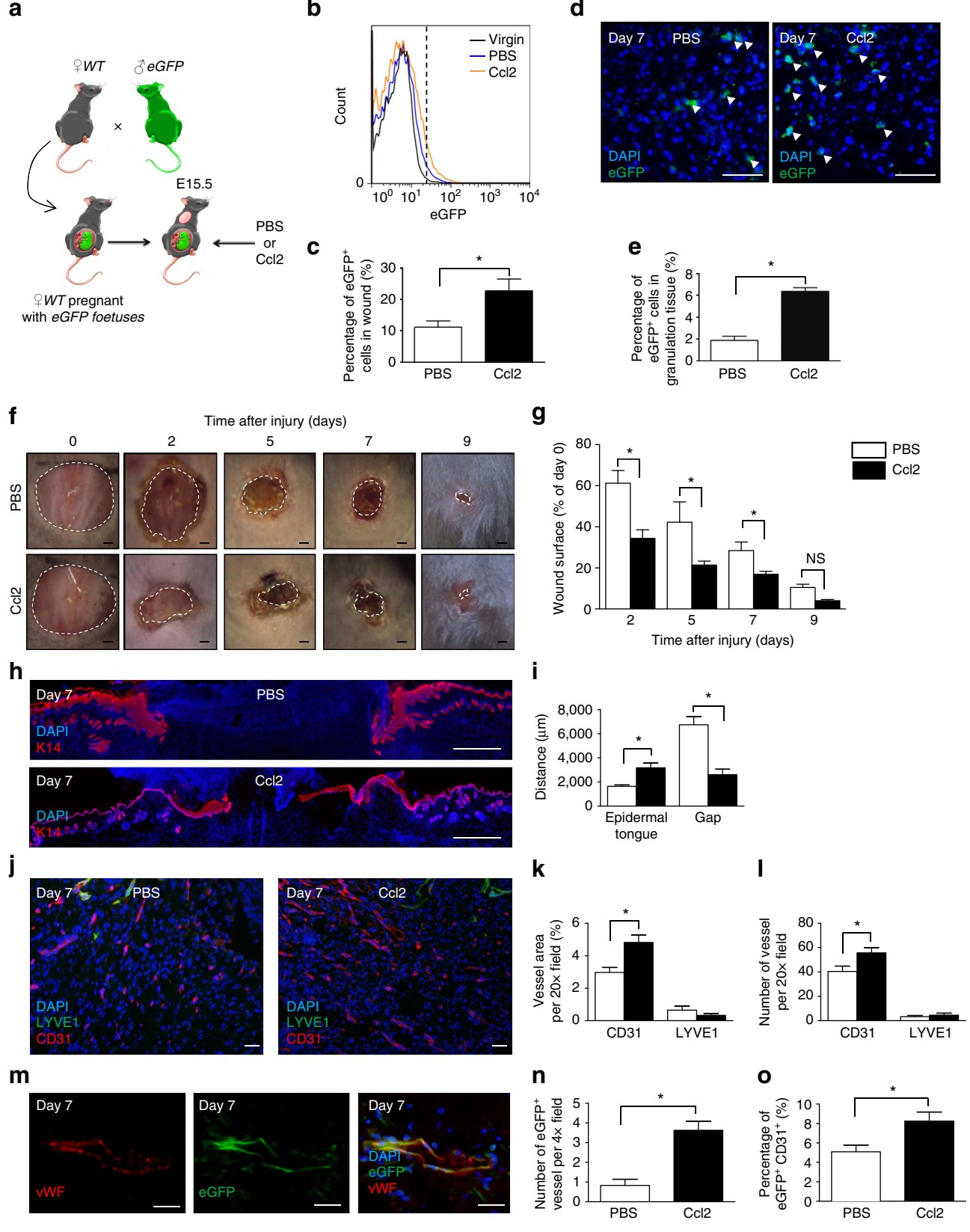

receiving PBS (Fig. 2d,e). The enhanced mobilization of eGFP+ cells in wounds upon Ccl2 administration was also found at days 3 and 5 (Supplementary Fig. 4). Therefore, Ccl2 is able to recruit FMCs to wound sites throughout the cutaneous healing process.

**Ccl2 improves wound healing by enhancing neovascularization.** We then analysed the effect of local Ccl2 treatment on maternal wound healing. On days 3, 5 and 7 after wounding, the neo-epidermal tongue, which reflects the healing without

contraction interference, showed a 43.44, 65.76 and 51.36% increased re-epithelialization, respectively, after Ccl2 injection as compared to controls at correspondent dates (Fig. 2h,i and Supplementary Figs 7a,b and 8a,b). Consistent with these findings, the non-healed area was decreased on days 2, 5 and 7 in wounds treated with Ccl2 (Fig. 2f,g). Cell proliferation in the epidermis and dermis areas increased on days 3, 5 and 7 in mice treated with Ccl2 (Supplementary Figs 3b,c; 5c,d and 6c,d). Of note, on day 9, there was no more difference in Ccl2- or PBS-treated mice in neo-epidermal tongue (Supplementary Fig. 7a,b), non-healed area (Fig. 2f,g) and epidermal cell proliferation (Supplementary Fig. 7c,d). Neovascularization, as evaluated by determining total (adult and fetal) $CD31^+$ vessel density and the expression of the *VEGF-A*, *VEGFR1* and *VEGFR2* genes, also increased in the Ccl2-injected wounds at all time points (Fig. 2j–l and Supplementary Figs 3h; 5e,f; 6e,f and 7e,f). In accordance, FACS analysis of total $CD31^+$ endothelial cells showed a twofold increase upon Ccl2 injections (Supplementary Fig. 3j). On the contrary, Ccl2 administration had no effect on lymphatic angiogenesis (Fig. 2j–l and Supplementary Figs 3i; 5e,f; 6e,f and 7e,f). Nassar *et al.*[13] reported the formation of blood vessels by FMCs in maternal wounds. We also identified Von Willebrand Factor (vWF)-positive blood vessels formed entirely or partly by $eGFP^+$ FMCs in Ccl2-treated mice (Fig. 2m). In these mice, there was also a higher number of fetal-derived vessels as well as $CD31^+$ $eGFP^+$ fetal endothelial cells measured by FACS than in those treated with PBS (Fig. 2n,o). Of note, wound inflammation, as assessed by determining the number of $GR-1^+$ neutrophils and $F4/80^+$ macrophages in granulation tissue, displayed no difference between mice treated with Ccl2 and those receiving PBS, including at early time points (Supplementary Figs 3d–g; 5g–j and 6g–j). We therefore conclude that Ccl2 improves healing by promoting the neovascularization of maternal wounds and the formation of fetal cell-derived vessels.

To further determine whether Ccl2 improves skin healing through the effects of FMCs or directly affect wound closure, we analysed the wound healing upon Ccl2 in virgin mice. In these, Ccl2 and PBS injections had similar effects on all parameters: wound surface area (Supplementary Fig. 8a,b), neoepidermal tongue (Supplementary Fig. 8c,d), proliferation (Supplementary Fig. 8e,f), angiogenesis and lymphangiogenesis (Supplementary Fig. 8g–k), and inflammation (Supplementary Fig. 9a–d). Therefore, the presence of FMCs is mandatory to promote maternal angiogenesis and wound healing through Ccl2.

**The mobilization of FMCs through Ccl2 is mediated by Ccr2.** At this stage, it was important to demonstrate whether the fetal cell signalling was dependent on Ccr2. To answer this question,

we analysed virgin female $Ccr2^{KO/KO}$ mice, female $Ccr2^{KO/KO}$ mice mated with $eGFP^+$ males and female $Ccr2^{KO/KO}$ mice mated with $eGFP^{KI}$ $Ccr2^{KO}$ males (Fig. 3a,c–e). When $Ccr2^{KO/KO}$ female mice bear $Ccr2^{KO/KO}$ foetuses, Ccl2 injections do not enhance wound healing at any day (Fig. 3c,d). In contrast, when $Ccr2^{KO/KO}$ female mice bear $Ccr2^{WT/KO}$ foetuses, Ccl2 decreases wounded area by 49.08, 28.96 and 57.58% at days 2, 5 and 7, respectively (Fig. 3e,f). Interestingly, this ratio is similar to the ratio we found with Ccl2 in WT mice. In addition, only $Ccr2^{KO/KO}$ mice bearing $Ccr2^{WT/KO}$ foetuses displayed an increase in fetal cell infiltrate in granulation tissue upon Ccl2 local injections (Fig. 3g,h). Finally, the virgin $Ccr2^{KO/KO}$ mice did not show any change when treated with Ccl2 (Fig. 3a,b). Therefore, all these data demonstrate that Ccl2 enhances wound healing through Ccr2-dependent fetal cell recruitment to wound bed.

**A specific subpopulation of FMCs responds to Ccl2.** We further investigated the subpopulations of fetal cells in the peripheral blood during wound healing. In wounded virgin mice, the number of total myeloid progenitor cells (MPCs), defined as $CD11b^+$ $CD34^+$ $CD31^+$ cells, persisted at the same level as in unwounded conditions during the first 2 days, then decreased at day 3 (Fig. 4a,b). In pregnant mice, the number of maternal MPCs, defined as $eGFP^-$ $CD11b^+$ $CD34^+$ $CD31^+$ cells, displayed the same pattern as the virgin mice (Fig. 4c,d). By contrast, the number of fetal MPCs, defined as $eGFP^+$ $CD11b^+$ $CD34^+$ $CD31^+$ cells, peaked on the day after wounding (Fig. 3e,f). We also studied maternal ($eGFP^-$) and fetal ($eGFP^+$) EPCs, defined as $CD11b^-$ $CD34^+$ $CD31^+$ cells. Both populations displayed a late increase on day 3 after wounding (Supplementary Fig. 10a–c). These observations demonstrate that only the fetal MPCs population expands early, 1 day after maternal wounding.

We further investigated the response of this fetal population of MPCs upon Ccl2 administration. Following the injection of Ccl2, the number of fetal MPCs decreased in the peripheral blood (Fig. 4g,h) but increased in the wound tissue (Fig. 4i,j). By contrast, the number of maternal MPCs in both blood and wound tissue was unaffected by Ccl2 (Fig. 4g–j). Meanwhile, Ccl2 did not modify the level of maternal and fetal EPCs in the blood and wound groups (Supplementary Fig. 10d,e). These results suggest that Ccl2 specifically recruits fetal MPCs from the blood to the wound during the healing process.

**Foetal MPCs play dual role in wound angiogenesis.** We investigated the fate of fetal MPCs *in vivo*, by purifying this population from the peripheral blood of pregnant mice with $eGFP^+$ foetuses on day 1 after wounding (Fig. 5a). As a control, we purified $eGFP^+$ adult MPCs from the peripheral blood of

---

**Figure 2 | Ccl2 recuits FMCs to maternal wounds and improve skin wound healing in pregnant mice.** (**a**) Experimental design: an 8 mm wound was created in pregnant female mice carrying $eGFP^+$ foetuses. We injected Ccl2 or PBS into the wound immediately and 2 days after skin excision. (**b**) FACS analysis showed that there were significantly larger numbers of $eGFP^+$ cells in the wounds of pregnant mice treated with Ccl2 than in those treated with PBS ($n = 3$). (**c**) Quantification of $eGFP^+$ cells in wounds on day 7 for pregnant mice receiving injections of PBS or Ccl2 ($n = 3$). (**d**) Wound sections from pregnant mice carrying $eGFP^+$ foetuses after the injection of PBS or Ccl2. Representative micrographs of the spontaneous fluorescence of $eGFP^+$ (green) cells in granulation tissue, indicated by white arrowheads. Scale bars: 50 μm. (**e**) Quantification of $eGFP^+$ cells in sections of wounds from pregnant mice carrying $eGFP^+$ foetuses after the injection of PBS or Ccl2 ($n = 5$). (**f**) Time course of skin wound healing. Scale bars: 1 mm (**g**) Planimetry of the wound area relative to the initial wound area at various time points ($n = 5$). (**h**) Anti-K14 (red) labelling of neoepidermal tongues and gaps in the wound. Scale bars: 1 mm. (**i**) Measurement of neoepidermal tongues and gaps at wound sites ($n = 5$). (**j**) Dual labelling for CD31 (red) and LYVE1 (green) in the wound. Scale bars: 50 μm. (**k**) Quantification of the relative vessel area per $20 \times$ field by fluorescence densitometry ($n = 4$). (**l**) Quantification of the number of vessel types per $20 \times$ field ($n = 4$). (**m**) Representative micrograph of a wound section labelled for vWF (red) and displaying spontaneous eGFP fluorescence (green), demonstrating the recruitment of FMCs to wound sites to form a vessel wall. Scale bars: 50 μm. (**n**) Quantification of $vWF^+$ $eGFP^+$ double-positive vessels in $4 \times$ field ($n = 4$). (**o**) FACS analysis showed that there were significantly larger numbers of $eGFP^+$ $CD31^+$ cells in the wounds of pregnant mice treated with Ccl2 than in those treated with PBS ($n = 3$). Student's *t*-test, $*P < 0.05$; mean ± s.e.m.

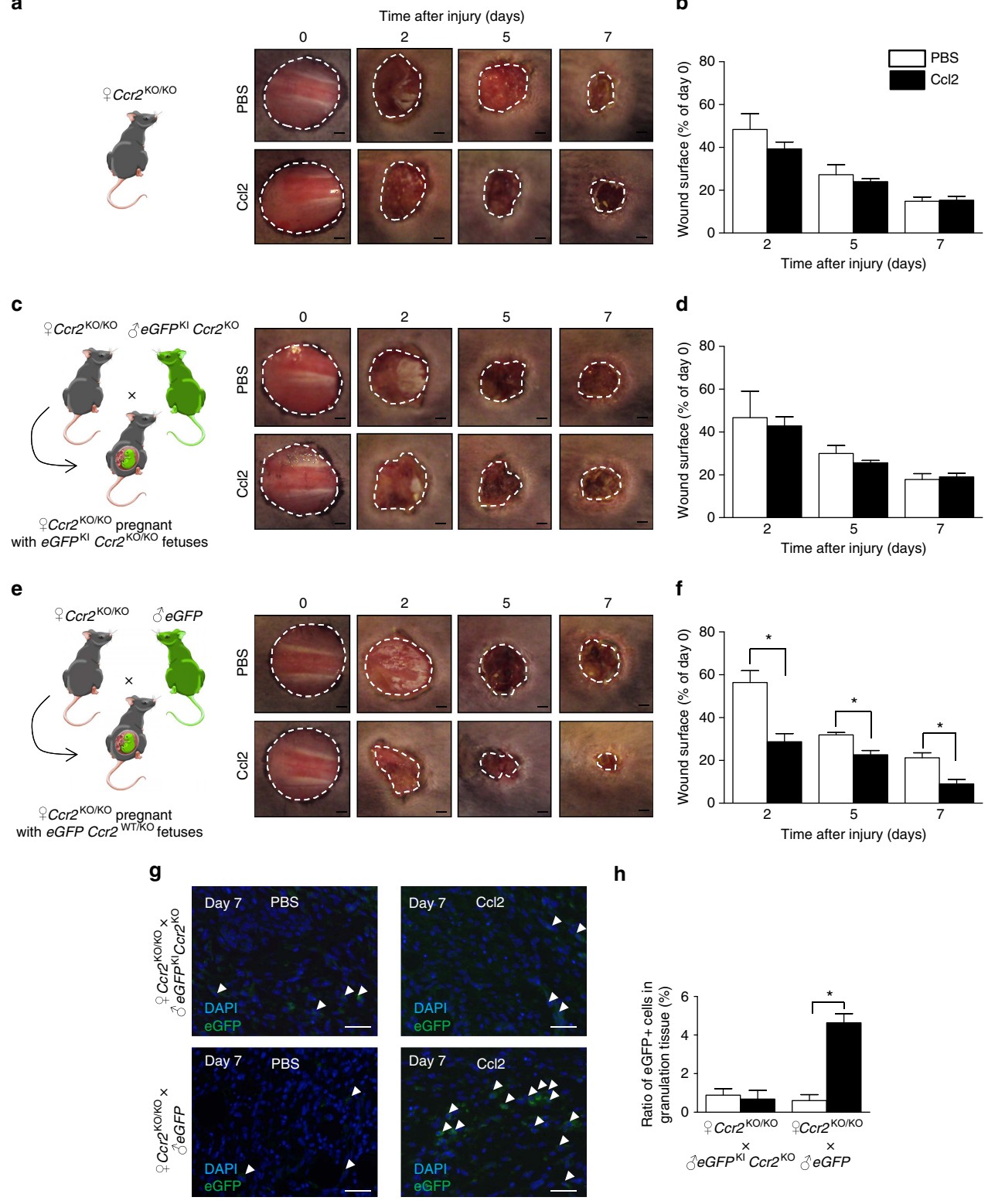

**Figure 3 | FMCs are recruited to maternal wounds through Ccl2/Ccr2 pathway.** An 8 mm wound was created in pregnant female mice carrying eGFP[+] foetuses. We injected Ccl2 or PBS into the wound immediately and 2 days after skin excision. (**a**) Experimental design and time course of skin wound healing from $Ccr2^{KO/KO}$ virgin mice. (**b**) Planimetry of the wound area relative to the initial wound area at various time points from $Ccr2^{KO/KO}$ virgin mice ($n = 5$). (**c**) Experimental design and time course of skin wound healing from $Ccr2^{KO/KO}$ female mice mated with $eGFP^{KI} Ccr2^{KO}$ male mice. (**d**) Planimetry of the wound area relative to the initial wound area at various time points from $Ccr2^{KO/KO}$ female mice mated with $eGFP^{KI} Ccr2^{KO}$ male mice ($n = 5$). (**e**) Experimental design and time course of skin wound healing from $Ccr2^{KO/KO}$ female mice mated with $eGFP$ male mice. Scale bars, 1 mm. (**f**) Planimetry of the wound area relative to the initial wound area at various time points ($n = 5$) from $Ccr2^{KO/KO}$ female mice mated with $eGFP$ male mice. (**g**) Representative micrographs of the spontaneous fluorescence of eGFP[+] (green) cells in granulation tissue indicated by white arrowheads. Scale bars, 50 μm. (**h**) Quantifications of eGFP[+] cells in sections of wounds from pregnant mice after injections of PBS or Ccl2 ($n = 3$). Student's t-test, *$P < 0.05$; mean ± s.e.m.

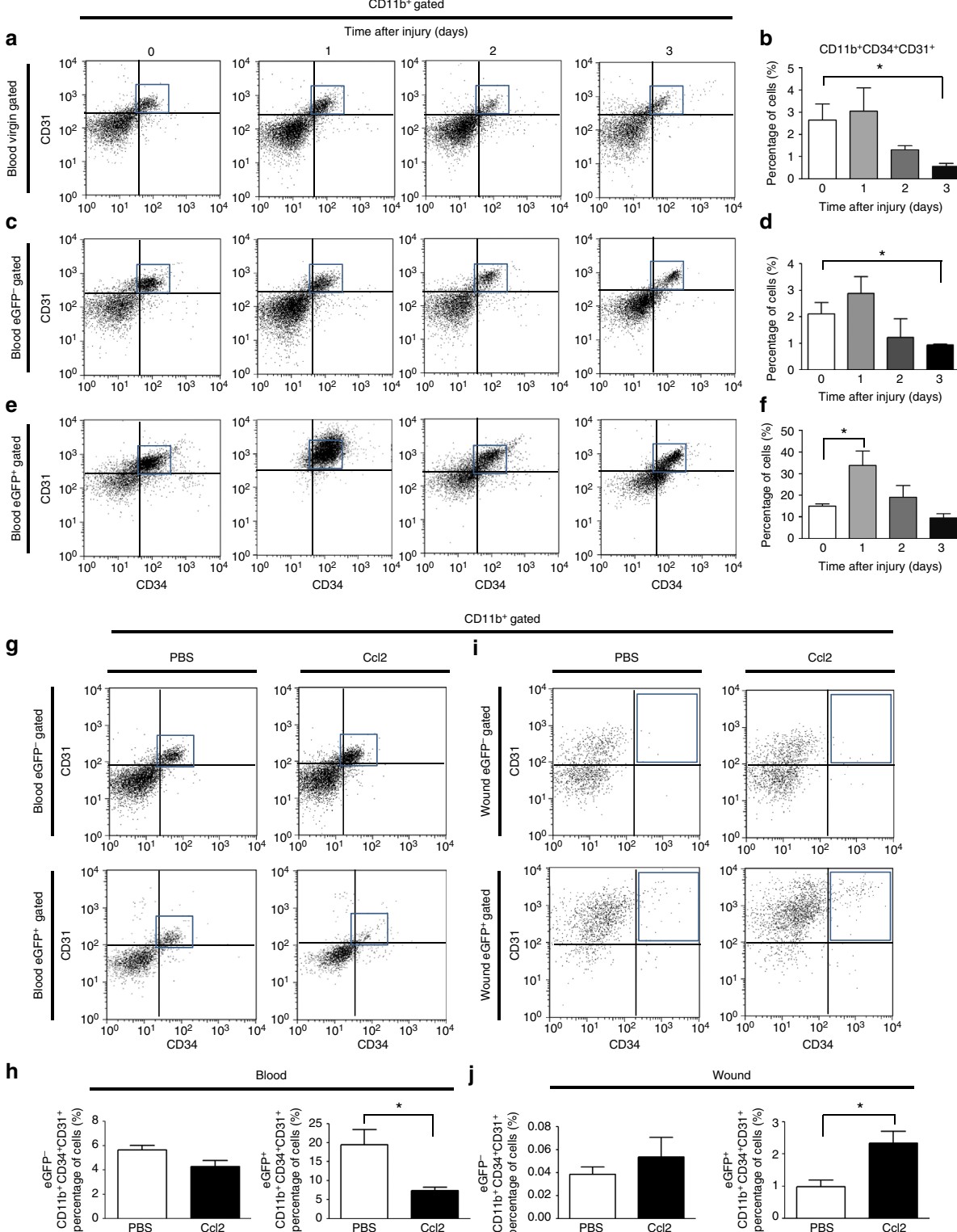

**Figure 4 | Ccl2 recruits fetal MPCs to wound.** Peripheral mononuclear blood cells (PBMCs) were collected from pregnant female mice carrying eGFP+ foetuses or from virgin control mice with and without wounds, on days 0–3 for FACS analysis. (**a,b**) Representative FACS results for CD34/CD31 staining in the CD11b+ gate for virgin mice, with quantification ($n = 3$). (**c,d**) Representative FACS results for CD34/CD31 staining in the CD11b+ gate in maternal cells (eGFP−) from pregnant mice, with quantification ($n = 3$). (**e,f**) Representative FACS results for CD34/CD31 staining in the CD11b+ gate for fetal cells (eGFP+) from pregnant mice ($n = 3$). (**b,d,f**) Percentage of CD11b+ CD34+ CD31+ cells. Pregnant female mice carrying eGFP+ foetuses were wounded and PBS or Ccl2 was injected into the wound immediately and 2 days after wounding. PBMCs and wounds were collected 7 days later for FACS analysis. (**g,h**) PBMCs and (**i,j**) wound tissues were analysed to determine the levels of maternal CD11b+ CD34+ CD31+ MPCs (eGFP− gate) and fetal CD11b+ CD34+ CD31+ MPCs (eGFP+ gate) present after the administration of PBS or Ccl2. (**g,i**) Representative FACS results for CD34/CD31 staining in the CD11b+/eGFP+/− gate. (**h,j**) Percentage of CD11b+ CD34+ CD31+ cells in the eGFP+/− gate ($n = 4$). Student's t-test, *$P < 0.05$; mean ± s.e.m.

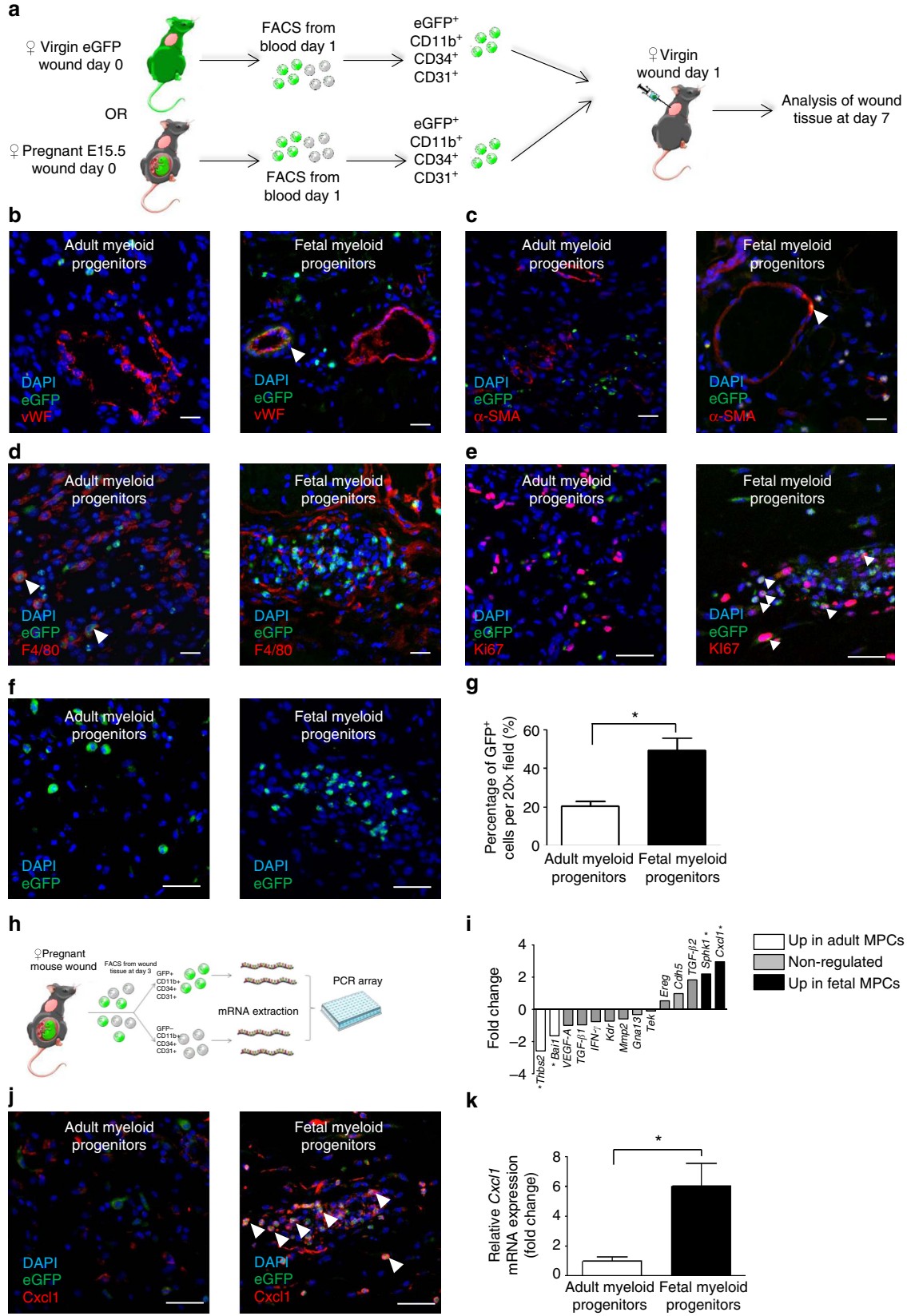

virgin wounded transgenic eGFP+ female mice (Fig. 5a). The cells were injected, on day 1, into the wounds of WT C57BL/6 virgin mice. At day 7, wounds injected with eGFP+ fetal MPCs contained vWF+ endothelial cells and were able to form a full vessel (Fig. 5b and Supplementary Fig. 11). In addition, some eGFP+ fetal MPCs-derived cells expressed smooth muscle cell actin (α-SMA) (Fig. 5c) but never macrophage markers (F4/80) (Fig. 5d). By contrast, following the injection of eGFP+ adult

MPCs into wounds, no endothelial or myofibroblastic eGFP+ cells were detected (Fig. 5b,c). Thus only fetal MPCs have the specific properties required for differentiation into endothelial and mural cells in maternal wounds.

We also investigated how fetal MPCs enhance angiogenesis in maternal wounds. We sorted fetal and maternal MPCs from wounded tissues of mice pregnant with eGFP+ foetuses on day 3 after wounding (Fig. 5h). We studied the expression profiles of genes associated with angiogenesis pathways by PCR array analysis (Fig. 5i,k). Five mRNA transcripts were upregulated in the fetal MPCs, including *Cxcl1*, *Sphk* and *TGF-β2*, *Cxcl1* being the highest increased one. Immunofluorescence analysis on wound sections into which fetal or adult MPCs were injected showed that only injected fetal MPCs in mice expressed *Cxcl1* (Fig. 5j). Foetal MPCs expressed lower levels of *Thbs2* and *Bai1*, two angiogenesis inhibitors (Supplementary Data 2).

**Foetal MPCs form proliferative clusters and express Ccr2.** The question of the clonal origin of the fetal population has repeatedly been raised and is beyond the scope of this article. We nevertheless examined wounds from virgin adult mice into which either adult or fetal MPCs had been injected. At day 7, wound sections into which we had injected adult MPCs contained isolated cells adult MPCs (Fig. 5e,f), whereas those injected with fetal cells contained clusters of eGFP+ Ki67 cells (Fig. 5e–g). Interestingl,y nearly all fetal but <1% of the adult MPCs expressed Ccr2 (Supplementary Fig. 12a–c). Furthermore, using FACS analysis performed on the blood of pregnant mice carrying eGFP+ foetuses, Ccr2 was expressed by 85.77 ± 1.048% of fetal MPCs and by only 0.313 ± 0.128% of their maternal counterparts (Supplementary Fig. 12e,f).

**Ccl2 rescues delayed wound healing in postpartum mice.** When present, wound-healing disorders affect adult females years after delivery. Since fetal cells persist in maternal BM throughout life[1], we investigated the effects of Ccl2 in a delayed wound-healing model, in the early and late postpartum periods. Two weeks or 6 months after delivery, females that had given birth to eGFP+ pups were treated daily with a topical application of 0.05% clobetasol on the lower back for 12 days. At that stage, the skin was atrophic. Excisional skin wounds were made in the clobetasol-treated areas. Ccl2 or PBS was injected into the wounds on days 0 and 2 after wounding (Fig. 6a). Ccl2 injections improved wound closure kinetic, neoepidermal tongues, epidermal and dermal cell proliferation and blood vessel angiogenesis in these mice (Fig. 6b–k). By contrast, lymphangiogenesis, as measured by LYVE1+ lymphatic vessel density and *VEGF-C* and *VEGFR3* gene expression, and inflammation, as measured by the number of GR1+ and F4/80+ cells in granulation tissue, did not change after Ccl2 injections (Fig. 6h–k and Supplementary Fig. 13a–e). Importantly,

Ccl2 recruited more fetal cells to the wound bed than PBS, as demonstrated by FACS analysis (Fig. 6l,m). Therefore, Ccl2 recruits FMCs and improves delayed wound healing in the early and late postpartum in mice (Supplementary Fig. 14a–h). Finally, Ccl2 treatment did not induce any systemic effect, with no impact on blood cells count, glycaemia and gamma-glutamyltransferase dosage (Supplementary Table 1).

## Discussion

We and others have shown that fetal progenitor cells are involved in maternal repair, including wound-healing processes, as well as in the liver, thyroid, myocardial infarction and brain excitotoxic lesions in humans and/or mouse models[8,10–13]. Here we purified fetal cells specifically involved in wound healing and determined their transcriptional profiles. These cells appeared to overexpress chemokine receptor Ccr2. We found that local injections of physiological doses of Ccl2, a Ccr2 ligand, improved skin healing only in pregnant and postpartum mice (Fig. 7). These data provide the first demonstration that a specific pathway involving physiological doses of Ccl2 acts exclusively through the mobilization of FMCs. Moreover; we have been able to show that Ccl2's ability to recruit FMCs in maternal wound healing was dependent upon Ccr2 expression on fetal cells. Previous studies have shown that, even at higher doses, Ccl2 has no effect on normal wound healing in virgin mice[30]. Ccl2 injections enhance wound healing in *db/db* mice, in which Ccl2 secretion is impaired[31]. Ccl2 plays a role in tissue repair, because complete Ccl2 deficiency results in an impairment of myeloid cell recruitment and skin healing[32]. Thus our data show for the first time the selective recruitment of fetal progenitor cells through low doses of Ccl2 into maternal injured tissue. This translates into enhanced maternal healing both during pregnancy and much later after delivery, in conditions of normal or delayed skin healing.

One key question concerns the advantages associated with the recruitment of fetal rather than adult cells. We identified the fetal cells mobilized by Ccl2 and recruited to wounds as CD11b+ CD34+ CD31+ cells. These proliferated in clusters and differentiated into vWF+ endothelial cells and α − SMA+ mural cells within the wound. By contrast, adult cells of the same phenotype never formed clusters; they had a low proliferative index and did not differentiate into vascular cells of any type. Our data indicate that such fetal progenitors, even when persisting in an adult, are more pluripotent and proliferative than adult progenitors in accordance with others[33]. Our results are consistent with the findings of several studies reporting an absence of adult EPCs[22,23], whereas such progenitors are found in the fetal compartment. Furthermore, fetal CD11b+ CD34+ CD31+ cells have an mRNA profile different from that of their adult counterparts, as they overproduce various proangiogenic molecules, including *Cxcl1*. Cxcl1/Cxcr2 signalling plays a

**Figure 5 | *In vivo* integration of transplanted fetal MPCs into the wound and fetal MPCs overproduce Cxcl1 in wounds.** (**a**) Experimental design: female CaG-eGFP mice or pregnant female mice carrying eGFP+ foetuses were wounded and eGFP+ CD11b+ CD34+ CD31+ myeloid progenitor or fetal MPCs were isolated from blood on the day after wounding. The recipient mice were normal virgin females with the same genetic background as the donor mice. We transplanted 1 × 10^4 adult MPCs into the wounds of recipient mice on day 1 after wounding, and the wound was harvested on day 7. Cryosections showing (**b**) anti-vWF (red) labelling, (**c**) anti-α-SMA (red) labelling, (**d**) anti-F4/80 (red) labelling, (**e**) anti-Ki67 (red) labelling and spontaneous eGFP (green) fluorescence. White arrowheads indicate colocalization. Cryosections showing (**f**) spontaneous eGFP (green) fluorescence after the transplantation of adult MPCs or fetal MPCs in the wounds of virgin mice. Scale bars: 50 μm. (**g**) Quantification of eGFP+ cells in the granulation tissue after the transplantation of MPCs or fetal MPCs. Experimental design: (**h**) pregnant female mice carrying eGFP+ foetuses were wounded and eGFP+ CD11b+ CD34+ CD31+ fetal MPCs and maternal eGFP− CD11b+ CD34+ CD31+ MPCs were isolated from wound tissue on day 3. We extracted mRNA from these cells and performed a high-throughput PCR array analysis. (**i**) PCR array analysis of angiogenesis-associated gene expression in fetal MPCs (black bars) and MPCs (white bars) from mice (n = 3). Cryosections showing (**j**) anti-Cxcl1 (red) and displaying spontaneous eGFP (green) fluorescence. Scale bars: 50 μm. (**k**) Quantitative RT–PCR validation of *Cxcl1* mRNA levels normalized against mRNA levels for *Gapdh* (n = 3). Student's *t*-test, *P < 0.05; mean ± s.e.m.

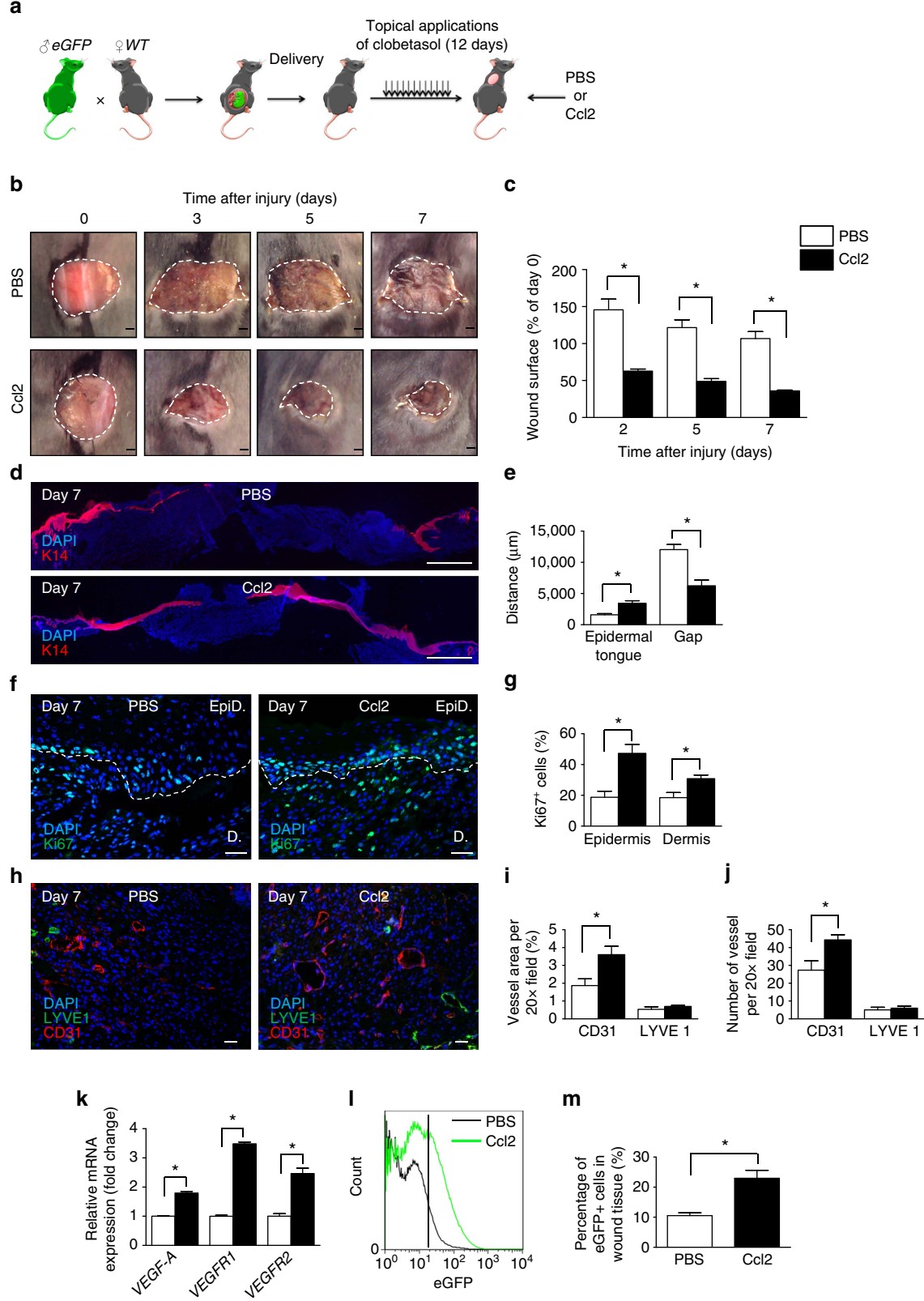

crucial role in neovascularization during skin repair[34]. Foetal progenitors persisting in the adult therefore reach maternal tissues and improve angiogenesis via the formation of new vessels derived from fetal cells and the stimulation of maternal angiogenesis.

The injection of as few as $10^4$ fetal cells led to clusters of proliferating fetal cells in the granulation tissue of injected wounds. Similarly, the maternal wounds displaying an improvement of healing after Ccl2 injection contained only $6.365 \pm 0.332\%$ fetal cells. Thus only very small numbers of fetal

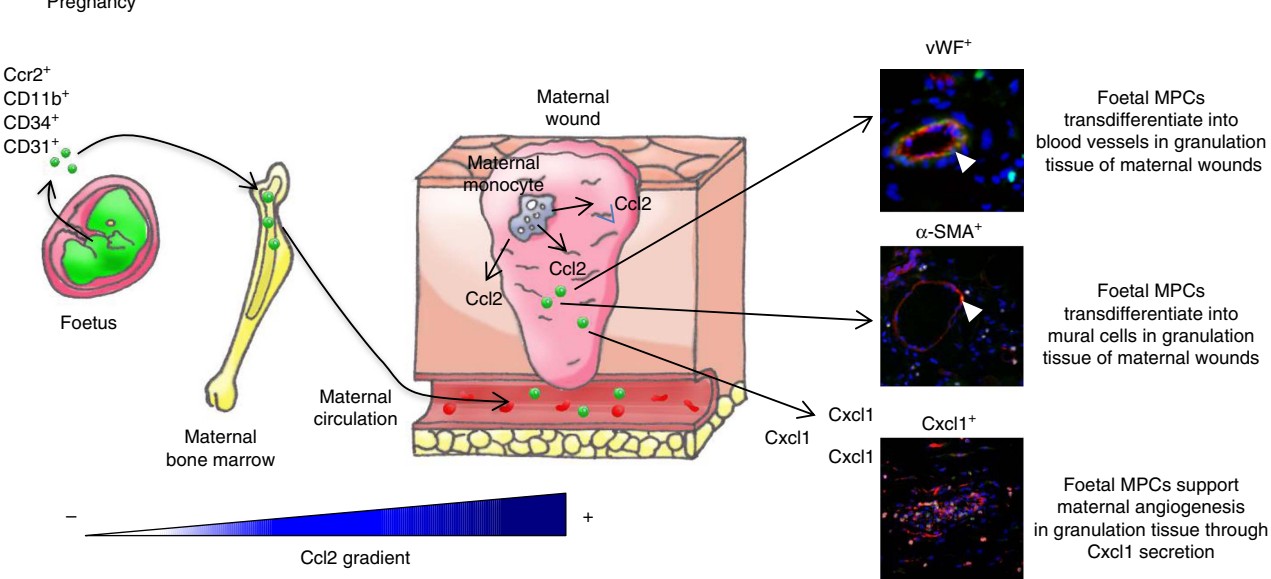

**Figure 7 | Mechanistic diagram of the role of fetal MPCs in maternal wound healing.** During pregnancy or postpartum, cutaneous wound monocytes and endothelial cells secrete Ccl2. This chemokine triggers the early recruitment of CD11b+ CD34+ CD31+ fetal cells from bone marrow to maternal wounds. Once these cells have reached the wound bed, they will differentiate into fetal-derived endothelial and mural cells. In addition, fetal cells secrete specific angiogenic signals such as Cxcl1 that increase maternal-derived angiogenesis. Overall, these mechanisms enhance maternal skin healing in pregnancy or postpartum.

cells are sufficient to improve both normal and delayed maternal wound healing. By contrast, stem cell therapy with adult cells requires the injection of millions of cells into wounds or myocardial infarcts[35–37]. Our results are in line with previous findings showing that, even in very low numbers, FMCs can rescue maternal defects[2,3]. This efficiency of small number of active fetal progenitors is consistent with the findings of Tamai et al.[38], who showed that Hmgb1 administration led to the recruitment of small numbers of BM Pdgfrα+ Lin− cells to skin injury sites, thereby improving delayed healing[38,39]. These findings can be extended to humans, as allogeneic BM transplantation rescues dystrophic epidermolysis bullosa in children through the migration of this population of cells to sites of skin injury and its differentiation into appropriate cell types[40].

In conclusion, this study lays the foundations for a new concept natural fetal stem cell therapy. We found that, due to the higher levels of Ccr2 expression on particular fetal progenitors, these cells could rescue delayed wound repair in parous mice and that this advantage persisted for a prolonged period after delivery. This strategy may provide an interesting new alternative to induced pluripotent stem cell, embryonic stem cell or allogeneic cell therapy. The mobilization of natural fetal stem cells results in the recruitment of cells well tolerated by the maternal immune

system[41]. Being semi-allogeneic, these cells should have beneficial effects in mothers with genetic or acquired diseases, such as diabetes or sickle cell disease[42]. In addition, fetal cells have several advantages over adult cells for this purpose, in terms of their pluripotency and proliferation capacity. Senescence levels are lower in fetal stem cells than in their adult counterparts[33], limiting the effects of aging. Finally, the mobilization of such cells avoids the need for various steps associated with risks of cell selection and amplification in vitro[35,43,44]. This approach could, theoretically, be extended to other organs, such as heart, liver or even the nervous system.

## Methods

**Mice.** Male mice transgenic for eGFP were obtained from Riken Laboratories (CD57BL/6-Tg (CAG-EGFP)1Obs/J), namely, eGFP and mated with wild-type (WT) 6-to-8-week-old C57BL/6 females from Harlan (Harlan). Female mice inactivated for Ccr2 (B6.129S4-Ccr2tm1Ifc/J), namely, Ccr2^KO/KO and male mice transgenic for eGFP gene inserted in Ccr2 gene (B6(C)-Ccr2tm1.1Cln/J), namely, eGFP^KI Ccr2^KO were obtained from Jackson Laboratory. Ccr2^KO/KO female were mated with male mice eGFP^KI Ccr2^KO. Animal experiments were performed according to experimental protocols following European Community Council guidelines and approved by our Institutional Animal Care and Use Committee (approval number 01161.02 and #8127).

**Flow cytometry.** The skin on the back of the mice was shaved and wounds were harvested and incubated overnight at 4 °C in 0.05% trypsin-EDTA (Invitrogen) for

**Figure 6 | Ccl2 improves delayed wound healing in postpartum conditions by recruiting FMCs.** (**a**) Experimental design: Each mouse received 12 daily topical applications of dermoval cream after the delivery. An 8 mm wound was created on the clobetasol-treated skin of the female mice postpartum (these mice had carried eGFP+ foetuses) and we injected PBS or Ccl2 into the lesion immediately and 2 days after skin excision. (**b**) Time course analysis of the healing of the excisional skin lesion; a representative image is shown. Scale bars: 1mm. (**c**) Planimetry of the wound area relative to the initial wound area at each time point (n = 5). (**d**) Anti-K14 (red) labelling of neoepidermal tongues and gaps in the wound on day 7. Scale bars: 1mm. (**e**) Measurement of neoepidermal tongues and gaps (n = 4). (**f**) Anti-Ki67 (green) labelling of the wound. Scale bars: 50 μm. (**g**) Quantification of Ki67+ cells in epidermal wound edges (EpiD) and the dermal granulation tissues (D) (n = 3). (**h**) Dual labelling for CD31 (red) and LYVE1 (green). Scale bars: 50 μm. (**i**) Quantification of relative vessel area per 20 × field by fluorescence densitometry (n = 3). (**j**) Quantification of the number of vessel types per 20 × field (n = 3). (**k**) Quantitative RT–PCR analysis of VEGF-A, VEGFR1 and VEGFR2 mRNA levels normalized against mRNA levels for Gapdh (n = 3). (**l**) FACS analysis demonstrated the presence of a significantly larger number of eGFP+ cells in the wounds of postpartum mice treated with Ccl2 than in those treated with PBS (n = 3). (**m**) Quantification of eGFP+ cells in the wounds of postpartum mice receiving injections of PBS or Ccl2 (n = 3). Student's t-test, *P < 0.05; mean ± s.e.m.

mechanical separation of the epidermis. Tissues were digested by incubation in collagenase IV for 60 min at 37 °C, with vortexing every 10 min, and the resulting suspension was filtered through a cell strainer with 100 μm pores and then a cell strainer with 40 μm pores (BD Pharmingen), to obtain a single-cell suspension. Blood was collected from the heart of the mouse and peripheral mononuclear blood cells were separated from erythrocytes and platelets by the Ficoll 1.088 method (Health Care). The cells were harvested and the resulting suspension was washed with PBS (Life Technologies) and filtered through a cell strainer with 40 μm pores (BD Pharmingen). Bones from the legs were collected and BMs were flushed, after sectioning the two heads of the bone, with 1 ml of PBS with a syringe 1 ml and a needle of 25 G. The resulting suspension was washed with PBS and filtered through a cell strainer with 40 μm pores (BD Pharmingen). The antibodies used for cytometry were CD34-eFluor660 (1:100; eBioscience), CD11b-PERCP-Cy5.5 (1:100; eBioscience), CD31-PE-Cy7 (1:100; eBioscience) and CCR2 (1:100; Santa Cruz) crossed with an anti-goat-Alexa 555 (1:1,000 Invitrogen). Flow cytometric data were acquired with a BD LSRII (BD Pharmingen) machine and sorting was performed on a MoFlo cell sorter (Beckman Coulter). The cells were then analysed with the FlowJo software (Treestar, San Carlos, CA).

In order to assess the specificity and sensitivity of FACS technique, we performed a dilution curve of eGFP$^+$ splenocytes in WT splenocytes ranging from 100% to 0.0001%. FACS analysis always showed the expected amount of eGFP cells, even in very diluted specimens. (Supplementary Fig. 2a,b). The correlation curve between the expected percentage of eGFP cells and the detected percentage on the standard curve was perfectly linear (correlation factor $R^2 = 1$). Furthermore, we checked the purity of the sorted eGFP$^+$ cells from the dilution curve using quantitative PCR that showed 92–100% purity in all samples except for the most diluted one that had 81% purity. ApoB was used as a reference gene (Supplementary Fig. 2c).

**Immunostaining.** We cut 5 μm sections of frozen tissue. These sections were incubated with cold acetone for permeabilization and then blocked by incubation with 2% BSA (Sigma-Aldrich). The primary antibodies used included rat anti-mouse CD31 (1:40; BD Biosciences), rabbit anti-mouse K14 (1:1,000; Covance), rabbit anti-mouse LYVE1 (1:200; Abcam), rat anti-mouse F4/80 (1:250; Abcam), rat anti-mouse GR-1 (1:250; eBiosciences) rabbit anti-mouse Ki67 (1:200; Abcam), goat anti-mouse Ccr2 (1:200; Santa Cruz Biotechnology), goat anti-mouse Ccl2 (1:200; Santa Cruz) and rabbit anti-mouse vWF (1:800; Abcam). For immuno-fluorescence, we used the following secondary antibodies: goat anti-rabbit IgG labelled with Cy3 or Alexa 488, donkey anti-rat IgG labelled with Alexa 488 or Cy3 and rabbit anti-goat Alexa 555 (1:1,000; Invitrogen). Slides were counterstained with 0.3 μg ml$^{-1}$ 4,6-diamidino-2-phenylindole (Sigma-Aldrich).

**Microscopic scoring and measurements.** We used a Nikon Eclipse 90i fluorescence microscope equipped with a Nikon DS-Fi1C digital camera (Nikon, Tokyo, Japan). For cell scoring, we took photographs of three different fields and counted the labelled cells by fluorescence densitometry, reporting the number of cells as a percentage of the total (assessed as the total number of nuclei). The mean percentage of labelled cells was calculated for each specimen. Measurements were made with the ImageJ software (NIH, Bethesda, MD).

**RNA extraction quantitative PCR and RNA sequencing.** Total RNA was extracted from cells or tissues with Trizol reagent, in accordance with the manufacturer's (Invitrogen) instructions. For low amount of cells, RNA was extracted with the NucleoSpin RNA XS Kit, in accordance with the manufacturer's (Macherey–Nagel) instructions. It was then reverse-transcribed with the iScript cDNA Synthesis Kit (Bio-Rad). The resulting cDNA was used for PCR with the SYBR-Green Master PCR Mix (Roche). PCR and data collection were performed on a LightCycler 480 (Roche). The levels of gene expression in the samples were normalized against that of the housekeeping gene (β-actin). All the primers were from Qiagen (QuantiTect).

RNA sequencing was performed on sorted fetal cells from the peripheral blood obtained from wounded and unwounded pregnant mice. Parameters used were 2 reads × 25 million fragments, paired-end 2 × 100 nt. Each sample was a pool of six mice, leading to >500 fetal cells per sample.

RNA sequencing analyses were performed at ICM, Hopital La Pitié Salpétrière, with an Illumina Hi-Seq device.

**Surgical wounds.** Mice were anaesthetized by the inhalation of 4.9% isoflurane delivered at a flow rate of 300 ml min$^{-1}$ in ambient air. The backs of the animals were shaved, and punch biopsy devices were used to create four 6 mm surgical wounds or a single 8 mm surgical wound. All tissues above the panniculus carnosus were excised. Wounds were left uncovered until harvesting. Standardized images of the wounds were obtained at various time points, with a Sony Cybershot 10.1-megapixel DSC-W180 digital camera (Sony, Tokyo, Japan). Wound tissues were harvested and either snap-frozen in liquid nitrogen or stored at −80 °C.

**Corticoid treatment.** Four days after delivery, mice were treated by the topical application of 200 μl clobetasol (Dermoval) per day onto shaved dorsal skin over a period of 10 or 12 days depending on age.

**Chemokine/cell injections.** Following the generation of an 8 mm wound, 100 μl of Ccl2 (Clinisciences, Nanterre, France) was injected into the four cardinal points of the wound bed on days 0 and 2, at a concentration of 0.5 ng μl$^{-1}$. We therefore injected 50 ng of Ccl2 after the surgery and 48 h later[26,27].

Alternatively, 10,000 cells were injected into the wound after FACS sorting and resuspension in PBS in accordance with the procedure used before, on day 1.

**Western blotting.** Wound samples were homogenized in RIPA buffer (Bio-Rad, Hercules, CA) supplemented with Complete Protease Inhibitor Cocktail (Roche) and centrifuged to obtain lysates. Equal amounts of extracted protein (20 μg) were subjected to SDS–polyacrylamide gel electrophoresis in a NuPAGE 4–12% Bis-Tris Gel (Novex, Invitrogen) and transferred to nitrocellulose membranes (GE Healthcare, Glattbrugg, Switzerland). The membranes were incubated with antibodies directed against β-actin (1:2,000; rabbit polyclonal; Cell Signaling) or against Ccr2 (1:500; goat polyclonal; Santa Cruz). They were then incubated with horseradish peroxidase-conjugated anti-rabbit antibody (1:2,500; Cell Signaling) or horseradish peroxidase-conjugated anti-goat antibody (1:5,000; Santa Cruz). Immune complexes were visualized with ECL prime (GE Healthcare) and signals were captured on ChemiDoc (Bio-Rad). Protein bands were analysed by densitometry with ImageJ. The uncropped Ccr2 membrane is shown in Supplementary Fig. 15.

**PCR array for chemokines and receptors and angiogenesis.** Changes in chemokines and receptors or angiogenesis levels were measured with an RT$^2$ Profiler PCR Array for mouse chemokines and receptors PAMM-022ZA or mouse angiogenesis PAMM-024ZR (Qiagen, Hilden, Germany). Total RNA was extracted from sorted eGFP$^+$ cells from BM. The expression of 86 cytokine and chemokine genes was analysed with the Lightcycler 1536 system (Roche Diagnostics, Mannheim, Germany), according to the manufacturer's instructions. The data obtained were analysed with the RT$^2$ Profiler PCR Array Data Analysis Template (Qiagen). Data were normalized against five housekeeping genes (Actb, B2m, Gapdh, Gusb, Hsp90ab1), and relative expression levels were calculated by the $2^{-\Delta\Delta Ct}$ method.

PCR array analyses were performed at the Hopital La Pitié Salpétrière ICM, with a LightCycler 1536 (Roche) and RT$^2$ Profiler PCR Arrays.

**Statistical analysis.** Statistical analysis was performed with the Graphpad Prism software. The results are expressed as means ± s.e.m. Pairs of groups were performed with unpaired, two-tailed Student's $t$-tests. Differences were considered statistically significant if $P < 0.05$.

**Data availability.** The authors declare that all data supporting the findings of this study are available within the article and its Supplementary Information files or from the corresponding author upon reasonable request.

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

## Acknowledgements

This work was supported by grants from Société Française de Dermatologie and AREMPH. Z.W. was supported with a Region Ile de France (DIM Stempole) post-doctoral funding. M.C. and M.S. received support from the French Ministry for Education (MEN). We also thank Annie Munier, Tatiana Ledent and Olivier Bregerie for their help in FACS and animal facilities.

## Author contributions

S.A. designed the study and supervised the research. M.C., M.S., M.J. and Z.W. performed all experiments. M.C., D.N. and Z.W. performed data analysis. M.C. drew all the schemes. M.C., D.N., Z.W. and S.A. wrote the manuscript.

## Additional information

**Competing interests:** A patent has been filed linked to this work, entitled 'METHODS AND PHARMACEUTICAL COMPOSITIONS FOR THE TREATMENT OF TISSUE LESIONS' (Deposit number: EP16305512.2, in May 2016), by authors S.A., M.C., D.N. and Z.W. All other authors declare no competing financial interests.

