## [Peer Review File · Nature Communications]

Reviewer #1 (Remarks to the Author)

A. Summary. The manuscript presents a large body of data to support the conclusion that CCL2/CCR2 signaling in female mice recruits fetal microchimeric cells that were implanted in bone marrow during a previous pregnancy and that the cells improve the healing of skin wounds. The conclusion extends fascinating previous observations about fetal microchimeric cells and may suggest new therapies with the cells or by mobilization of the cells.

B. Originality and interest: High because of the implications for new therapies.

C. Data and methodology: Several concerns. Fig. 1a shows an increase from 0.0001% to 0.0007% of fetal eGFP cells in bone marrow of the mice. My arithmetic says this is an increase from about 1 to 8 cells per million. If so, are these reproducible assays? Have the authors carried out standard curves and internal controls with spiking for the assays? Unfortunately, the Methods do not present any protocol for collection and processing of bone marrow. And why are the assays carried out only on day 1, 2 and 3? Wound healing is a dynamic process and a more detailed time course would be of great interest. But the next step is also of concern. If only 8 cells per million in bone marrow respond, are the data on RT-PCR assays of just 12 cytokine receptors in bone marrow (Fig. 1d and supplement) meaningful? There are also concerns about experiments in which CCL2 was injected into skin wounds. The protocols for the injections are not described, and there is no indication of volume or dose injected except for statement that a "physiological dose" was used. Without such information, how can other investigators reproduce the results? This is particularly troubling since injection of CCL2 or PBS is repeatedly used to generate much of the data.

D. Appropriate use of statistics and treatment of uncertainties: Generally acceptable with reservations above.

E. Conclusions: robustness, validity, reliability. The major conclusions may well be correct and have very important implications. However, as indicated in C., major parts of the paper do not meet the new standards for robustness, validity and reliability now being called for by many scientific groups and the NIH grant review administration. It is somewhat unfair to abruptly impose new scientific standards on a manuscript such as the present one that incorporates an immense amount of work by a large group of creative and dedicated scientists. But is there a fair way to begin?

F. Suggested improvements: Provide responses to questions and details requested above. Delete data that do not fully meet the new standards.

G. References: Appropriate.

H. Clarity and context: Extremely well addressed in the manuscript.

Reviewer #2 (Remarks to the Author)

In this manuscript, Castela et al. proposed a novel concept of the possible use of fetal stem cells for therapy. The authors present the mechanism through which fetal microchimeric cells (FMCs) are recruited to the wound area to promote the wound healing process after wounding in maternal mice. The authors propose that this process occurs via the CCR2/CCL2 signal between the wound area and CCR2 expressing FMCs. Addressing following concerns/questions will help improve the manuscript.

Although the authors suggested that CCL2 improves maternal wound healing, the wounds in both control and experimental groups did not display complete wound closure during the wound healing. Moreover, it is very difficult to determine whether the wound healing effect is significant in the presented figures (Especially, CCL group at 5 and 7 day in Fig. 2e). Mouse wounds heal mostly by contraction. It appears that the treatment is effective to promote wound contraction than healing. The authors should analyze the timing of completion of epithelialization and assess the final size of the wound scar area.

The authors described that fetal cells rapidly increased in the BM, blood and wound bed 1 day after wounding, and CCL2 was markedly overexpressed at 1 day after wounding (Fig. 1). However, the authors only analyzed the wound healing upon CCL2 at 7 day after wounding (Fig. 2, 4, and 6).

They need to further perform analysis at earlier times during wound healing

The functional role of this pathway is primarily tested via the injection of CCL2 to the wound area. It would be interesting to assess whether knocking out either CCL2 in the mother or fetal CCL2 would disrupt the recruitment of FMCs to the wound area, and ultimately disrupting the angiogenesis process or the wound healing process upon CCL2 addition.

Can injection of fetal MPCs on virgin mice have an effect on the wound healing process in wounded virgin mice upon or without treatment of CCL2? Or is it specific to maternal wounds?

The authors wrote...“but none of the adult MPCs expressed CCR2 (Supplementary Fig. 10a-c)” however there are CCR2+EGFP- cells present within the figure. The authors should modify the text accordingly.

In Fig 5e, there are some cells that express CXCL1 upon injection of adult MPCs, and the authors should adjust the text accordingly as they state that only mice receiving fetal cells express this.

Immunohistochemical data (Supplementary Fig. 4e) and immunoblot data (Supplementary Fig. 2a) need to accompany accurate quantitative analysis.

Reviewer #3 (Remarks to the Author)

A. Summary of key results

This manuscript describes a role for CCR2 expression on fetal microchimeric cells in maternal wound healing via increased production of CCL2 (the ligand for CCR2).

B. Originality and interest

Participation of these fetal derived cells in maternal wound healing has been previously described in murine models, but the requirement for CCR2/CCL2 is not previously described. Requirement for CCL2 for wound healing has also been previously described, but not a specific role in recruiting fetal microchimeric cells.

C. Data/methodology

Studies utilize tracking eGFP+ cells from fetuses in eGFP- maternal mice in healing wounds. Expression of CCR2 and CCL2 are determined quantitatively and data is supportive of conclusions.

D. Statistics

Differences are somewhat small for most figures, but statistical significance is appropriately indicated.

E. Robustness of conclusions

The conclusion that CCL2 in a wound increases recruitment of microchimeric fetal mesenchymal cells into maternal wounds and that expression of CCR2 is increased in these cells. The conclusion that fetal derived cells increase wound healing versus wound healing in virgin mice is well documented.

F. Suggested improvement

The studies provide evidence that increased CCL2 increases recruitment of the fetal cells to the wound, but not that this is necessary for this recruitment. The conclusion would be more impactful if a genetic model or another approach was used to substantiate the studies with CCL2 injection in the wound. It is possible that CCL2 attracts these cells, but is not necessary for recruitment of these cells under normal wounding conditions.

Increased expression of CCR2 in the fetal derived cells is similarly not functionally demonstrated to be necessary for recruitment of these cells to a wound. Other approaches should be used to determine the functional role of CCR2 in this process.

G. References are fine

H. Clarity of writing is adequate.

Response to Reviewers

We thank the reviewers for their constructive comments. We have addressed most, if not all of their comments and suggestions, performed many new experiments, which we hope addressed their criticisms and improved the quality and the impact of our manuscript

In summary,

1. In order to show the specificity and sensitivity of our assays for detecting eGFP fetal cells, we provide now standard curves of eGFP detection by flow cytometry and by quantitative PCR on serial dilutions ranging from 10% of eGFP in WT cells to 0.0001%.
2. We analyzed more time points of wound healing and provide now a fully detailed time course analysis of macroscopic wound closure as well as histological parameters of wound healing. We included early (days 3 and 5) and late (day 9) time points in the experiments shown in the article.
3. We performed transcriptional profiling by RNA sequencing of sorted fetal cells from wounded versus unwounded pregnant mice. Fetal cells from 5 mice in each condition were pooled to allow the analysis of a larger number of cells and the samples was performed in duplicates. This transcriptional analysis allowed us to identify a number of surface receptors overexpressed when maternal mice are wounded. It also confirms the overexpression of *Ccr2* in fetal cells from wounded mice that we previously showed in the high throughput PCR assay.
4. We now provide functional demonstration of the implication of the *Ccl2/Ccr2* pathway in the recruitment of fetal cells in maternal wound beds and their effect on maternal wound healing by using mice models with *Ccr2* knock out. For this purpose, we mated *Ccr2*^{KO/KO} females with eGFP^{KI} *Ccr2*^{KO} males -or with eGFP WT males- in order to allow gestations bearing fetuses with or without *Ccr2* knock-out and expressing GFP. Wound healing experiments were performed on pregnant mice from these mating's, with *Ccl2* of PBS injections in the wound beds. The results nicely show that when *Ccr2* is knocked out on fetal cells, *Ccl2* does not enhance wound healing and wounds from these mice show minimal fetal cell recruitment. This provides functional evidence that *Ccl2* enhances wound healing through recruitment of *Ccr2* expressing fetal cells.
5. We enhanced the quality of the immunofluorescence pictures as well as all the main and supplementary figures.

We hope that you will find our new manuscript suitable for publication in Nature Communications.

Please find below point-by-point answers on the reviewers' comments.

Reviewer #1

Several concerns. Fig. 1a shows an increase from 0.0001% to 0.0007% of fetal eGFP cells in bone marrow of the mice. My arithmetic says this is an increase from about 1 to 8 cells per million. If so, are these reproducible assays? Have the authors carried out standard curves and internal controls with spiking for the assays?

We agree with the reviewer that this is an important point. The proportion of fetal cells in various maternal compartments is low. Nevertheless, our assays for detecting GFP fetal cells, whether by flow cytometry, quantitative PCR or immunofluorescence are highly sensitive and very reproducible. In order to better demonstrate this, we provide a standard curve done with serial dilutions of eGFP cells in WT cells, isolated from the spleen of eGFP or WT mice ranging from 10% eGFP cells to 0.0001% GFP. Positive control was 100% GFP cells and negative control was 100% WT cells. As shown in the flow cytometry dot plots in the new Supplementary figure 2, FACS always detected the expected amount of eGFP cells, in all samples including very diluted specimens. Correlation curve between the real percentage of eGFP cells and the detected percentage on the standard curve was perfectly linear (Correlation factor $R^2 = 1$).

Besides, we studied the purity of eGFP+ cell sorting using 2 techniques. We performed quantitative PCR for eGFP on the serial dilutions and on FACS sorted eGFP+ cells from all the serial dilution samples. As shown in new supplementary figure 2c and d, 92 to 100% of sorted eGFP+ cells by FACS were indeed eGFP+ through qPCR except in the most diluted sample where the concentration of GFP+ cells by qPCR was 81%. In addition, we also did a second FACS analysis on sorted eGFP cells: 95% of the cells in the eGFP+ gate were again Dapi-eGFP+ (data not shown). All these informations have been included in the new MS page 4, Section Results, Paragraph Maternal skin wounding recruits FMCs through Ccr2 signaling, line 23-24 and page 16, Section Material and Methods, Paragraph Flow cytometry, line 4-13.

Besides, in the experiment show in figure 1a and b, which is the detection by flow cytometry of eGFP+ cells in the bone marrow and blood of pregnant mice at different time points after wounding, we added more replicates for every time point and additional late time points. The histograms in figure 1a show now data for n=7 mice at day 0, n=9 mice for day 1 n= 6 mice for day 2 and n=6 for day 3. The new histograms show the same kinetics with more analyzed mice adding to the reproducibility and consistency of our assays and data.

Unfortunately, the Methods do not present any protocol for collection and processing of bone marrow. And why are the assays carried out only on day 1, 2 and 3? Wound healing is a dynamic process and a more detailed time course would be of great interest.

As requested by the reviewer, we now added the detailed protocols that we followed for collection and processing of bone marrow (Section Material and Methods, Paragraph Flow cytometry, Page 15, line 20-24).

We also did later time points, namely days 5 (n=3 mice) and 9 (n = 3 mice), in the assay of detection of eGFP+ cells by flow cytometry in maternal blood, marrow and skin post wounding. Of note, numbers of fetal cell in these compartments at these later times points remained low. These results are in accordance - concerning the skin- with those from Seppanen et al. (Ref 24 of the MS). These data have been implemented in new Figure 1a.

But the next step is also of concern. If only 8 cells per million in bone marrow respond, are the data on RT-PCR assays of just 12 cytokine receptors in bone marrow (Fig. 1d and supplement) meaningful?

We thank the reviewer for this comment. For the transcriptional profiling of fetal cells in the bone marrow, we performed high throughput RT-PCR done on fetal cell isolated from the bone marrow of 12 pregnant mice: 6 wounded versus 6 unwounded. In each specimen, the number of sorted fetal cells was between 10 and 30 cells. A set of 90 genes corresponding to chemokines and chemokines receptors were screened in this assay. We kept for final analysis genes that showed expression in all analyzed specimens in both conditions. These correspond to the 13 transcripts shown in Fig. 1d. Amongst these, CCR2 demonstrated the highest fold change.

Moreover, in order to address the reviewer's concern, we performed RNA Sequencing (50 million reads paired, read length 100bp) on sorted fetal cells from the peripheral blood obtained from wounded (n = 2 samples) and unwounded (n = 2 samples) pregnant mice. We pooled circulating fetal cells from 5 mice in each sample, leading to more than 500 fetal cells/sample. We first considered only genes that displayed high expression (>100 count per million) in wounded mothers. Then we selected the genes that were overexpressed in circulating fetal cells from wounded mice as compared to unwounded mice (> 1.25 fold change). Using these thresholds, we retrieved a list of 558 genes, overexpressed in circulating fetal cells

from wounded mice. Among these genes, 16 corresponded to surface receptors and the only chemokine receptor was Ccr2, that was 1.44 higher in wounded vs unwounded conditions (Fig. 1e).

These data are now detailed page 5, Section Results, Paragraph Maternal skin wounding recruits FMCs through Ccr2 signaling, line 3-12 and shown in new Fig. 1e.

There are also concerns about experiments in which CCL2 was injected into skin wounds. The protocols for the injections are not described, and there is no indication of volume or dose injected except for statement that a "physiological dose" was used. Without such information, how can other investigators reproduce the results? This is particularly troubling since injection of CCL2 or PBS is repeated used to generate much of the data.

The protocols for dilutions and injections of CCL2 were given in the previous version (section Material Methods, Paragraph Chemokine/Cell injections page 18), but have been now put more clearly and with additional details in page 6, line 15 and Section Material and Methods, paragraph chemokine/cell injection page 18 of the new version. The dose has been derived from previous papers that studied CCL2 and skin wound healing. In one paper, (Wood et al, Ref 27 in MS) the authors injected 60 ng in the granulation tissue of wounds on diabetic mice. In another paper (Dipietro et al, Ref 26 in MS), the authors injected a range of doses from 50 to 200 ng in wound bed with no observed toxicity. Finally, Maharshak et al., (Inflamm Bowel Dis 2010;16:1496–1504) injected daily what they describe as various low dosages of CCL2 (30, 60, 120, 180 ng/mL) in intra-peritoneum. Based on these papers, we chose to inject 50 ng at day 0 and day 2 of skin wounds in the wound beds only.

Reviewer #2

Although the authors suggested that CCL2 improves maternal wound healing, the wounds in both control and experimental groups did not display complete wound closure during the wound healing. Moreover, it is very difficult to determine whether the wound healing effect is significant in the presented figures (Especially, CCL group at 5 and 7 day in Fig. 2e). Mouse wounds heals mostly by contraction. It appears that the treatment is effective to promote wound contraction than healing. The authors should analyze the timing of completion of epithelialization and assess the final size of the wound scar area.

We agree with the Reviewer about the importance of contraction in mice skin wound healing. However, we have also used a widely accepted parameter of healing, namely the neo-epidermal tongue, that we have measured on medial sections at various time points. As seen in the Fig 2, 6 and new supplementary Fig 5, 6, 7 and 8 the length of the neo-epidermis -that is independent of contraction- is significantly increased when CCL2 was injected at D0 and D2 to pregnant or post partum mice. Of note, epidermal as well as dermal proliferation measured through Ki67 labeling was also enhanced upon CCL2. We believe therefore that these results clearly indicate that CCL2 improves healing *per se*. These informations have been highlighted in the results of the new MS page 7, Section Results, Paragraph CCL2 administration improves wound healing by enhancing neovascularization in pregnant but not in virgin mice line 4-17.

In addition, as requested we have performed analyses at additional time points after wounding, early (days 3 and 5) and late (day 9), to have a complete time course analysis of various parameters of wound healing. This result has been put in new supplementary Figure 5, 6, 7.

The authors described that fetal cells rapidly increased in the BM, blood and wound bed 1 day after wounding, and CCL2 was markedly overexpressed at 1 day after wounding (Fig. 1). However, the authors only analyzed the wound healing upon CCL2 at 7 day after wounding (Fig. 2, 4, and 6). They need to further perform analysis at earlier times during wound healing

As requested, we have performed new experiments and provide now a full time course analysis at days 3, 5, 7 and 9 after skin wounding. CCL2 nicely improved wound healing in pregnant mice. Interestingly, the analysis of specimens did not show any changes in macrophage recruitment upon Ccl2 injections, including at early time points. This indicates that at low doses, CCL2 does not trigger monocyte recruitment but in contrast acts on fetal progenitor mobilization. These results have been added to the new supplementary Figures 5, 6 and 7 and discussed in page 7-8, paragraph "CCL2 administration improves wound healing by enhancing neovascularization in pregnant but not in virgin mice".

The functional role of this pathway is primarily tested via the injection of CCL2 to the wound area. It would be interesting to assess whether knocking out either CCL2 in the mother or fetal CCL2 would disrupt the recruitment of FMCs to the wound area, and ultimately disrupting the angiogenesis process or the wound healing process upon CCL2 addition.

We thank the reviewer for his comment. We have carefully considered the reviewer's requests about a more accurate demonstration of the implication of the Ccl2/Ccr2 pathway in fetal progenitor's mobilization. Wound maternal cells secrete Ccl2 that act on Ccr2+ fetal cells. It appeared therefore in our view more demonstrative to analyze the consequences of knocking down Ccr2 on fetal cells.

For this purpose, we mated Ccr2^{KO/KO} females with eGFP^{KI} Ccr2^{KO} males or with eGFP WT males. This led to gestations of females bearing fetuses with or without Ccr2 knock-out but expressing GFP. Wound healing experiments were performed on pregnant mice from these matings, with CCL2 or PBS injections in the wound beds. The results clearly show that when Ccr2 is knocked out on fetal cells, Ccl2 does not enhance wound healing and that wounds from these mice show low fetal cell recruitment. These results provide functional evidence that Ccl2 enhances wound healing through recruitment of Ccr2 expressing fetal cells. These results have been detailed in page 8-9, Section Results, New Paragraph The mobilization of FMCs through Ccl2 is mediated through Ccr2 and are shown in new Figure 3.

Can injection of fetal MPCs on virgin mice have an effect on the wound healing process in wounded virgin mice upon or without treatment of CCL2? Or is it specific to maternal wounds?

We agree with the reviewer that the injection of fetal microchimeric cells sorted from pregnant wounded mice into wound beds of virgin mice would assess whether their effect on wound healing is specific to maternal wounds. Nevertheless, this experiment does not really reflect any natural physiological condition as fetal cells are only present in a microchimeric state in pregnant and parous mice. Besides, the strategy that we have adopted is to study a way to trigger a natural phenomenon rather than perform a cellular therapy.

The authors wrote..."but none of the adult MPCs expressed CCR2 (Supplementary Fig. 10a-c)" however there are CCR2+EGFP- cells present within the figure. The authors should modify the text accordingly.

We thank the reviewer for his comment. We agree with the reviewer that there are few adult MPCs that show CCR2 expression. The text of the new version has been modified accordingly (Results, paragraph Fetal myeloid progenitor cells form proliferative cluster and express Ccr2, page 11, line 12).

In Fig 5e, there are some cells that express CXCL1 upon injection of adult MPCs, and the authors should adjust the text accordingly as they state that only mice receiving fetal cells express this.

We thank the reviewer for his comment. Indeed there are cells expressing CXCL1 found in wound beds after injection of adult MPCs although they are much fewer than in wounds injected with fetal MPCs. This is now mentioned in the new version (Results, paragraph "Fetal myeloid progenitor cells play dual role in wound angiogenesis", page 11, line 1).

Immunohistochemical data (Supplementary Fig. 4e) and immunoblot data (Supplementary Fig. 2a) need to accompany accurate quantitative analysis.

In Supplementary figure 8e, we now provide representative immunofluorescence data that accompany corresponding quantitative analysis presented in the histogram in Supplementary figure 8f.

Similarly, the immunoblot data in Figure 1k are now accompanied by quantification presented in the histogram in Figure 1k.

--

Reviewer #3

The studies provide evidence that increased CCL2 increases recruitment of the fetal cells to the wound, but not that this is necessary for this recruitment. **The conclusion would be more impactful if a genetic model or another approach was used to substantiate the studies with CCL2 injection in the wound. It is possible that CCL2 attracts these cells, but is not necessary for recruitment of these cells under normal wounding conditions.**

Increased expression of CCR2 in the fetal derived cells is similarly not functionally demonstrated to be necessary for recruitment of these cells to a wound. Other approaches should be used to determine the functional role of CCR2 in this process.

We thank the reviewer for his comment that converges with one comment of reviewer 2. As detailed above, to provide functional demonstration on the role of Ccl2/Ccr2 pathway in attracting fetal cells in maternal skin wounds, we mated Ccr2^{KO/KO} females with eGFP^{K1}Ccr2^{KO} males or with eGFP WT males. This led to gestations of females bearing fetuses with or without Ccr2 knock-out but expressing GFP. Wound healing experiments were performed on pregnant mice from these matings, with CCL2 or PBS injections in the wound beds. The results clearly show that when Ccr2 is knocked out on fetal cells, Ccl2 does not enhance wound healing and that wounds from these mice show low fetal cell recruitment.

These results, are detailed in page 8-9, Section Results, New Paragraph The mobilization of FMCs through Ccl2 is mediated through Ccr2 and in new Figure 3 provides functional evidence that Ccl2 fetal cell triggering acts through recruitment of Ccr2 expressing fetal cells.

Reviewer #1 (Remarks to the Author)

The authors have adequately responded to my criticisms and suggestions.

Reviewer #2 (Remarks to the Author)

The authors made efforts to address questions and added additional data. I do not have additional comments.

Reviewer #3 (Remarks to the Author)

The authors have satisfactorily addressed the concerns at the time of the prior review.

Many additional studies were performed which enhance the significance of the work and strengthen the conclusion.